# Isolation and Characterization of a Novel Phage Collection against Avian-Pathogenic *Escherichia coli*

Marianne Nicolas,[a] Angélina Trotereau,[a] Antoine Culot,[b] Arshnee Moodley,[c] Robert Atterbury,[d] Jeroen Wagemans,[e] Rob Lavigne,[e] Philippe Velge,[a] Catherine Schouler[a]

aINRAE, Université de Tours, ISP, Nouzilly, France
bRime Bioinformatics SAS, Palaiseau, France
cDepartment of Veterinary and Animal Sciences, University of Copenhagen, Frederiksberg, Denmark
dSchool of Veterinary Medicine and Science, University of Nottingham, Leicestershire, United Kingdom
eDepartment of Biosystems, Laboratory of Gene Technology, KU Leuven, Leuven, Belgium

**ABSTRACT** The increase in antibiotic-resistant avian-pathogenic *Escherichia coli* (APEC), the causative agent of colibacillosis in poultry, warrants urgent research and the development of alternative therapies. This study describes the isolation and characterization of 19 genetically diverse, lytic coliphages, 8 of which were tested in combination for their efficacy in controlling *in ovo* APEC infections. Genome homology analysis revealed that the phages belong to nine different genera, one of them being a novel genus (*Nouzillyvirus*). One phage, REC, was derived from a recombination event between two *Phapecoctavirus* phages (ESCO5 and ESCO37) isolated in this study. Twenty-six of the 30 APEC strains tested were lysed by at least one phage. Phages exhibited varying infectious capacities, with narrow to broad host ranges. The broad host range of some phages could be partially explained by the presence of receptor-binding protein carrying a polysaccharidase domain. To demonstrate their therapeutic potential, a phage cocktail consisting of eight phages belonging to eight different genera was tested against BEN4358, an APEC O2 strain. *In vitro*, this phage cocktail fully inhibited the growth of BEN4358. In a chicken lethality embryo assay, the phage cocktail enabled 90% of phage-treated embryos to survive infection with BEN4358, compared with 0% of nontreated embryos, indicating that these novel phages are good candidates to successfully treat colibacillosis in poultry.

**IMPORTANCE** Colibacillosis, the most common bacterial disease affecting poultry, is mainly treated by antibiotics. Due to the increased prevalence of multidrug-resistant avian-pathogenic *Escherichia coli*, there is an urgent need to assess the efficacy of alternatives to antibiotherapy, such as phage therapy. Here, we have isolated and characterized 19 coliphages that belong to nine phage genera. We showed that a combination of 8 of these phages was efficacious *in vitro* to control the growth of a clinical isolate of *E. coli*. Used *in ovo*, this phage combination allowed embryos to survive APEC infection. Thus, this phage combination represents a promising treatment for avian colibacillosis.

**KEYWORDS** *Escherichia coli*, bacteriophage therapy, genome analysis

Avian-pathogenic *Escherichia coli* (APEC) are responsible for the main extra-intestinal bacterial disease in poultry, colibacillosis, and the intestine constitutes a reservoir for the bacteria. Colibacillosis can affect all avian species, at all ages and in all types of poultry production. Indeed, APEC is responsible for a variety of syndromes, such as embryo mortality, omphalitis and yolk sac infection, salpingitis, swollen head syndrome, and respiratory tract infection which evolves into systemic diseases like perihepatitis, airsacculitis, and pericarditis (1). Several *E. coli* vaccines are available for avian

Address correspondence to Catherine Schouler, catherine.schouler@inrae.fr.

The authors declare no conflict of interest.

colibacillosis, but control of the disease mainly depends on the use of antibiotics (1). Increasing antibiotic resistance in APEC (2) warrants alternative control approaches, such as phage therapy.

Bacteriophages (phages) are viruses which specifically infect bacteria and represent the most abundant biological entity on the planet (3). Phages possess many advantages over antibiotics: they are usually easily isolated, have few—if any—side effects, and are specific to their bacterial hosts, which helps prevent the secondary infections or dysbiosis which may characterize the use of broad-spectrum antibiotics (4). In poultry, several studies have estimated the efficacy of phage therapy in colibacillosis treatment. When inoculated intramuscularly simultaneously with the pathogen and at a multiplicity of infection (MOI) of 1, phage R (genus *Vectrevirus*, family *Autographiviridae*), targeting the K1 capsule, allowed 100% of chickens to survive the disease (5). In addition, an inoculation of $10^6$ PFU of phage R, two days before infection, produced a significant degree of protection (only 14% mortality versus 100% for controls without phage) (5). Similarly, SPR02 phage (unknown genus) injected simultaneously with the pathogen and at an MOI of 1,000 enabled all chickens to survive (6). Finally, chickens infected and treated with TM3 phage (unknown genus) showed decreased mortality when the phage was administered intramuscularly (26.5% mortality versus 46.6% in untreated controls) (7).

The high specificity of phages is a major advantage because they specifically target a host and have no or minor impact on the microbiota. Conversely, this specificity can limit the therapeutic spectrum of the phage. Moreover, treatment with only one phage could increase the risk of selecting a phage-resistant bacterium. Thus, the combination of phages in cocktails allows broader coverage of different target bacterial strains and/or genera and can be used to prevent the emergence of phage-resistant clones if several phages target the same bacterial strain (8). The design of a phage cocktail involves the isolation of lytic phages from various environments (rivers, sewage, feces), allowing an important selection of taxonomically distinct phages which target different bacterial receptors, have a wide host range, and can overcome bacterial antiphage defense systems. Phage libraries already exist against *E. coli*, with descriptions of their sources, host ranges, and genome characteristics, e.g., the presence of lysogenic genes or toxins (9). However, few studies have investigated the impact of a phage cocktail against APEC strains. A cocktail consisting of six phages (EW2, TB49, AB27, KRA2, and TriM of unknown genus and the *Tequatrovirus* G28) inhibited growth and biofilm formation by some APEC strains (10). Moreover, another cocktail of four phages showed therapeutic protection with a 30% improvement in chicken survival and a 100-fold decrease in the total number of *E. coli* in the lungs (7). However, no studies have combined both genomic characterization of a collection of phages against APEC strains with an *in vivo* evaluation of their therapeutic potential to control avian colibacillosis.

This study aims (i) to isolate and characterize coliphages from various environmental samples which can multiply on avian-pathogenic *E. coli* strains, and (ii) to assess *in ovo* the efficacy of a phage cocktail at reducing embryo mortality induced by an APEC strain. Nineteen phages were characterized based on their morphology, taxonomy, genome, and host range. Among these, vB_EcoS_ESCO41 shares only 78% identity (92% coverage) with a phage in the database, leading to the definition of the new phage genus *Nouzillyvirus* in the family *Drexlerviridae*. The isolated phages belong to nine genera. The 19 coliphages exhibit varying infectious capacity, with narrow to broad host ranges. The most infectious phages belong to three genera: *Tequatrovirus*, *Nonagvirus*, and *Phapecoctavirus*. The *Nonagvirus* ESCO3 and two *Phapecoctavirus* (ESCO5 and REC) encoded a receptor-binding protein (RBP) containing a polysaccharidase domain, which could explain their greater infectivity. A phage cocktail consisting of 8 phages from eight genera was able to protect 90% of chicken embryos from APEC infection.

## RESULTS AND DISCUSSION

**Isolated coliphages belong to nine genera.** Forty-two phages were isolated from various environmental samples, including river and pond water, horse dung, and avian

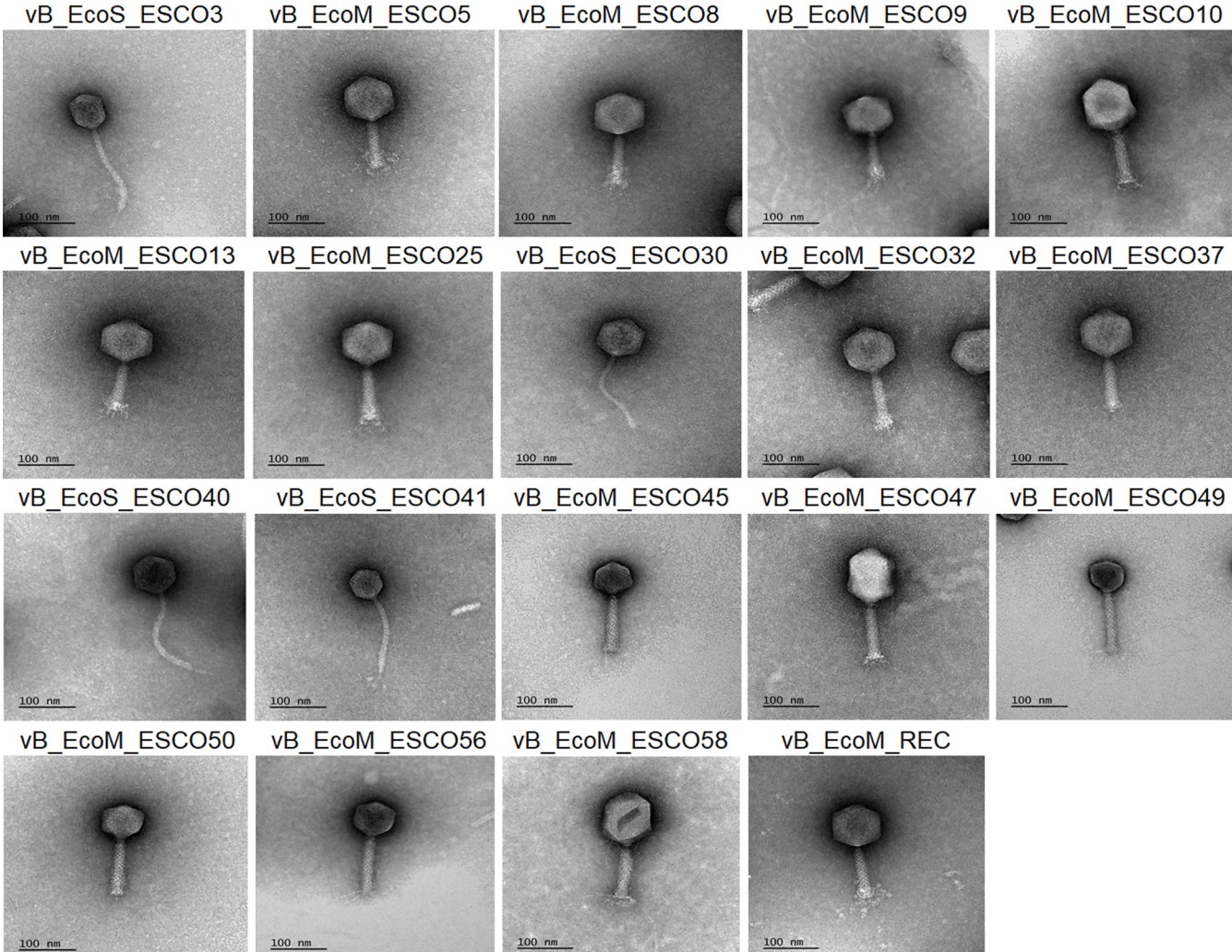

**FIG 1** Transmission electron micrographs of the coliphages. Purified phage particles were negatively stained with 1% (wt/vol) uranyl acetate and visualized by IBiSA electronic microcopy platform (Tours, France). Scale bars (nm) are included for each picture.

cecal content. According to HincII restriction fragment length polymorphism analysis, the 39 phages were clustered in 15 groups (Fig. S1 in the supplemental material). The genomes of three phages (ESCO10, ESCO47, and ESCO58) were not restricted by HincII. These latter phages and 1 phage per RFLP group were amplified, purified, and sequenced. The morphology of the 18 purified phages was determined by transmission electron microscopy (TEM): 14 phages have a myovirus morphology with a typical icosahedral head and a short contractile tail, and 4 phages have a siphovirus morphology with an icosahedral head and a long noncontractile tail (Fig. 1, Table S1).

The 18 phages belonged to nine distinct genera, as determined by homology at the nucleotide level in the NCBI database and confirmed by VIRIDIC analysis: *Justusliebigvirus*, *Phapecoctavirus*, *Felixounavirus*, *Dhakavirus*, *Mosigvirus*, *Tequatrovirus*, *Nonagvirus*, *Tequintavirus*, and *Nouzillyvirus* (Table 1, Fig. S2). All isolated phages were classified as *Gammaproteobacteria* host group, which is the Enterobacteria class. Based on the ESCO41 genome, the new genus *Nouzillyvirus* was defined by the International Committee on Taxonomy of Viruses (ICTV) in 2020 (https://ictv.global/taxonomy/taxondetails?taxnode_id=202108172), ESCO41 being the type species of this genus. To date, no phage sharing more than 80% coverage with the ESCO41 genome has been described (February 2023, NCBI database).

A phylogenetic tree of the 19 coliphages of this study and all phages belonging to the nine genera showed that genera belonging to the same subfamily were closely related (*Justusliebigvirus* and *Phapecoctavirus* of the subfamily *Stephanstirmvirinae*;

**TABLE 1** Phage taxonomy and similarity to closest known phage

| Phage | Taxonomy based on genome | | | Most homologous phage | | | |
| | Class/family | Subfamily | Genus | Name | Identity (%) | Coverage (%) | Accession no. |
|---|---|---|---|---|---|---|---|
| vB_EcoM_ESCO8 | *Caudoviricetes* | *Stephanstirmvirinae* | *Justusliebigvirus* | *Escherichia* phage outra | 99.63 | 96 | MN850645 |
| vB_EcoM_ESCO9 | *Caudoviricetes* | *Stephanstirmvirinae* | *Justusliebigvirus* | *Escherichia* phage alia | 98.27 | 97 | NC_052655 |
| vB_EcoM_ESCO32 | *Caudoviricetes* | *Stephanstirmvirinae* | *Justusliebigvirus* | *Escherichia* phage outra | 97.86 | 97 | MN850645 |
| vB_EcoM_ESCO5 | *Caudoviricetes* | *Stephanstirmvirinae* | *Phapecoctavirus* | *Escherichia* phage phAPEC8 | 97.72 | 92 | JX561091 |
| vB_EcoM_ESCO13 | *Caudoviricetes* | *Stephanstirmvirinae* | *Phapecoctavirus* | *Escherichia* phage phAPEC8_ev052 | 99.21 | 98 | LR597654 |
| vB_EcoM_ESCO25 | *Caudoviricetes* | *Stephanstirmvirinae* | *Phapecoctavirus* | *Escherichia* phage phAPEC8_ev052 | 98.91 | 99 | LR597654 |
| vB_EcoM_ESCO37 | *Caudoviricetes* | *Stephanstirmvirinae* | *Phapecoctavirus* | *Escherichia* phage phAPEC8_ev052 | 98.27 | 98 | LR597654 |
| vB_EcoM_REC | *Caudoviricetes* | *Stephanstirmvirinae* | *Phapecoctavirus* | *Escherichia* phage phAPEC8 | 97.72 | 91 | JX561091 |
| vB_EcoM_ESCO45 | *Caudoviricetes* | *Ounavirinae* | *Felixounavirus* | *Escherichia* phage dune | 96.18 | 94 | MN850636 |
| vB_EcoM_ESCO49 | *Caudoviricetes* | *Ounavirinae* | *Felixounavirus* | *Escherichia* phage dune | 96.22 | 95 | MN850636 |
| vB_EcoM_ESCO50 | *Caudoviricetes* | *Ounavirinae* | *Felixounavirus* | *Escherichia* phage dune | 95.96 | 93 | MN850636 |
| vB_EcoM_ESCO56 | *Caudoviricetes* | *Ounavirinae* | *Felixounavirus* | *Escherichia* phage dune | 95.93 | 92 | MN850636 |
| vB_EcoM_ESCO10 | *Straboviridae* | *Tevenvirinae* | *Dhakavirus* | Enterobacteria phage vB_EcoM_VR5 | 93.56 | 88 | KP007359 |
| vB_EcoM_ESCO47 | *Straboviridae* | *Tevenvirinae* | *Mosigvirus* | *Escherichia* phage vB_EcoM_G2469 | 93.44 | 90 | MK327934 |
| vB_EcoM_ESCO58 | *Straboviridae* | *Tevenvirinae* | *Tequatrovirus* | *Escherichia* phage vB_EcoM_G4500 | 98.17 | 98 | MK327945 |
| vB_EcoS_ESCO3 | *Caudoviricetes* | *Queuovirinae* | *Nonagvirus* | *Escherichia* phage vB_EcoS_HdK1 | 96.22 | 92 | MK373794 |
| vB_EcoS_ESCO30 | *Demerecviridae* | *Markadamsvirinae* | *Tequintavirus* | *Escherichia* phage vB_EcoS_VAH1 | 93.87 | 85 | MK373792 |
| vB_EcoS_ESCO40 | *Demerecviridae* | *Markadamsvirinae* | *Tequintavirus* | Bacteriophage T5-like saus111K | 96.27 | 87 | MF431734 |
| vB_EcoS_ESCO41 | *Drexlerviridae* | | *Nouzillyvirus* | Phage vB_EcoM-p111 | 92 | 78 | OL449681 |

*Dhakavirus*, *Mosigvirus*, and *Tequatrovirus* of the subfamily *Tevenvirinae*) (Fig. 2). These observations were confirmed by the scored phylogenomic similarities calculated using VIRIDIC (Fig. S2). *Justusliebigvirus* phages shared in average 46.4% nucleotide identity with the *Phapecoctavirus* phages. *Mosigvirus* phage shared 48.8% nucleotide identity with *Dhakavirus* phage, and both shared 43.5 and 54.1% nucleotide identity, respectively, with the *Tequatrovirus* phage. The phylogenetic tree, in addition to the taxonomy, shows a high diversity among our isolated phages. Similar results have already been observed in the phage collection of Townsend et al. (11): 30 *Klebsiella* phages isolated from various water samples belonged to nine phylogenetically distinct genera. Except for the genus *Nouzillyvirus*, all phage genera identified in this collection have already been identified in several studies (12, 13).

The *Phapecoctavirus* genus was named by Korf et al. (14) based on phAPEC8, described in 2012 (15), which was isolated from a chicken water sample on an APEC strain. The 4 *Phapecoctavirus* phages were isolated from different samples, such as sewage water, horse dung, and avian cecal content (Table 2). The diversity of sources from which *Phapecoctavirus* phages are isolated has already been described in the *Phapecoctavirus* genomes available as of 2012: 9/16 from water samples, 2/16 from compost, 1/16 from human feces, 1/16 from farm chicken, and 3/16 from avian fecal content (16). This indicates that *Phapecoctavirus* phages can infect *E. coli* strains from different origins.

When *Phapecoctavirus* phages ESCO5 and ESCO37 were added to a culture of strain Cp6salp3, extended bacterial growth retardation was observed compared to bacterial growth in the presence of each phage individually; note that ESCO5 induces no bacterial growth retardation (Fig. S3A). Moreover, clearer and bigger plaques could be observed (Fig. S3B). The phage was amplified from one of these purified plaques and named REC.

**The genomic organization of phages from the same genus shows significant synteny.** The genomic features of the 19 phages are listed in Table 2. All the encoded proteins were present in databases, except one protein of the *Nouzillyvirus* phage ESCO41 (ESCO41_00041) (February 2023, NCBI). Because no integrase nor excisionase-encoding genes were found in the phage genomes, we hypothesized that the 19 phages have a virulent lifestyle.

Comparison of the 19 phages genome showed synteny between phages belonging

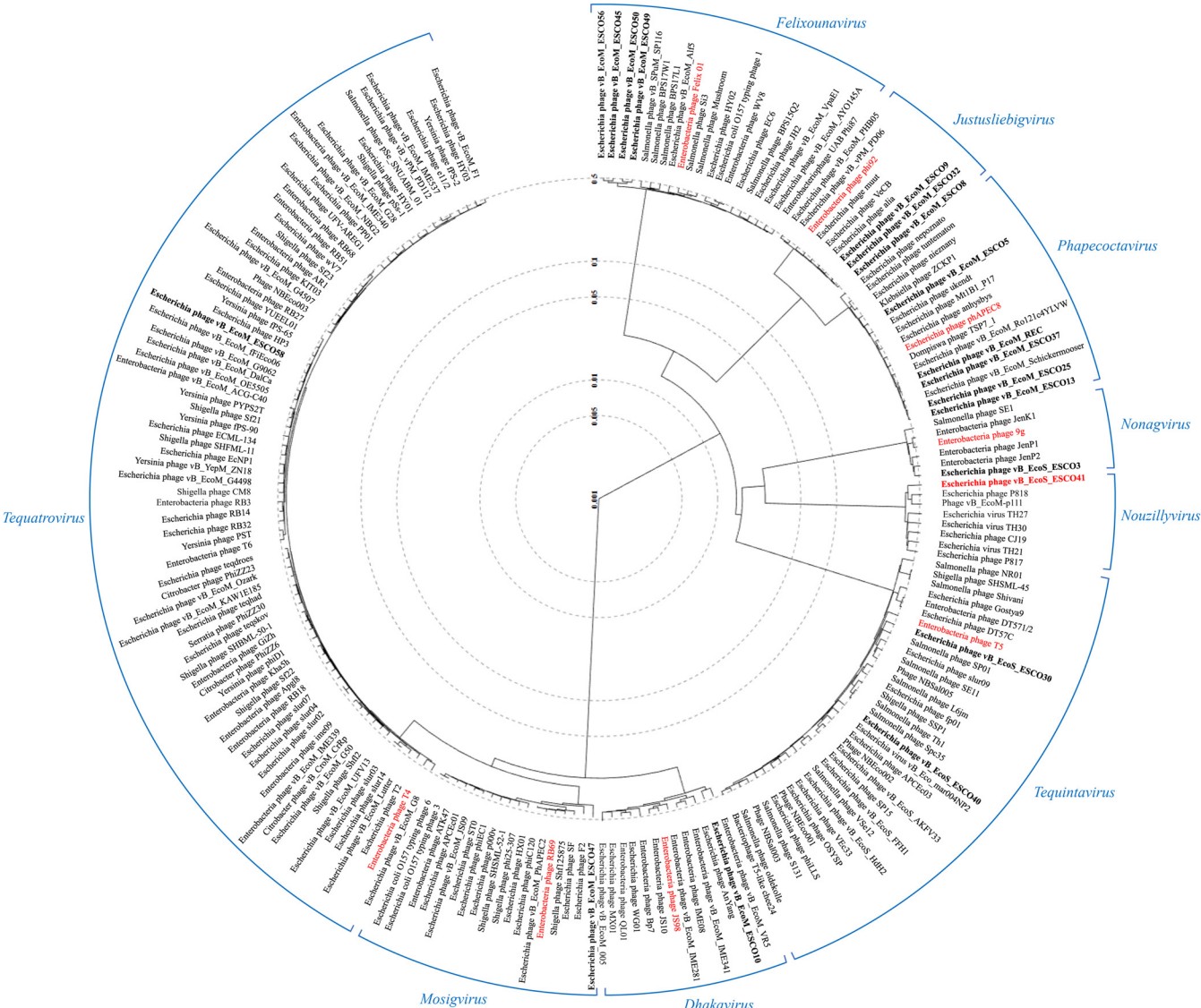

**FIG 2** Proteomic phylogenetic tree. Circular proteomic tree generated by VIPtree including the 19 coliphages and the 181 phage genomes of each genus represented in this study. Genera are indicated in blue. The 19 phages are shown in bold, and genus-defining phages are in red.

to the same genus (Fig. 3). However, differences in structural modules, specifically in the tail genes, which code the main determinants of host specificity, could be observed between phages within the same genus.

The three *Justusliebigvirus* phage genomes shared nearly 98% identity (>95% coverage; Fig. 3A). In the structure and packaging module, the main differences were that ESCO8 encoded an additional tail spike protein (ESCO8_00245) and ESCO9 has an additional endo-*N*-acetylneuraminidase gene (ESCO9_00183).

The five *Phapecoctavirus* phages shared 97% identity (>91% coverage; Fig. 3B). The differences between them were mainly genes of unknown function which were absent or replaced by other genes (Fig. 3B). ESCO37 was the only one that did not encode a putative glucose-1-phosphate thymidylyl-transferase and a putative dTDP-glucose 4,6-dehydratase, enzymes involved in the metabolism of nucleotide sugars. Comparison of the genomes clearly showed that REC is the result of homologous recombination between ESCO5 and ESCO37 when the two phages coinfect Cp6Salp3. Indeed, the REC genome is identical to the ESCO37 genome from 0 to 11,573 bp and from 117,476 to 143,665 bp, identical to the ESCO5 genome from 34,365 to 117,08 5bp, and identical to the genomes of both phages from 11,573 to 34,365 bp and from 117,085 to 117,476 bp

**TABLE 2** Origin and genomic features of the 19 isolated coliphages

| Phage | Origin | Genome size (bp) | GC content (%) | tRNA no. | Protein no. | GenBank accession no. |
|---|---|---|---|---|---|---|
| vB_EcoM_ESCO8 | Avian cecal content | 148,860 | 37.5 | 16 | 265 | OM386653 |
| vB_EcoM_ESCO9 | Avian cecal content | 145,392 | 37.5 | 15 | 255 | OM386654 |
| vB_EcoM_ESCO32 | Pond water | 147,304 | 37.4 | 15 | 259 | OM386658 |
| vB_EcoM_ESCO5 | Avian cecal content | 149,268 | 39.0 | 10 | 273 | KX664695 |
| vB_EcoM_ESCO13 | Sewage | 149,813 | 39.1 | 10 | 285 | KX552041 |
| vB_EcoM_ESCO25 | Sewage | 149,073 | 39.0 | 10 | 284 | OM386656 |
| vB_EcoM_ESCO37 | Horse dung | 149,194 | 39.9 | 10 | 290 | OM386659 |
| vB_EcoM_REC | Laboratory | 156,300 | 38.9 | 10 | 296 | OM386667 |
| vB_EcoM_ESCO45 | Horse dung | 82,361 | 39.0 | 18 | 130 | OM386661 |
| vB_EcoM_ESCO49 | Horse dung | 84,827 | 39.0 | 19 | 136 | OM386663 |
| vB_EcoM_ESCO50 | Horse dung | 85,919 | 38.9 | 15 | 133 | OM386664 |
| vB_EcoM_ESCO56 | Horse dung | 83,464 | 39.0 | 13 | 128 | OM386665 |
| vB_EcoS_ESCO3 | Avian cecal content | 59,956 | 43.8 | 0 | 84 | OM386652 |
| vB_EcoM_ESCO10 | Avian cecal content | 171,100 | 39.1 | 2 | 281 | OM386655 |
| vB_EcoS_ESCO30 | Avian cecal content | 112,115 | 38.9 | 22 | 183 | OM386657 |
| vB_EcoS_ESCO40 | River water | 108,755 | 39.2 | 18 | 188 | OM386660 |
| vB_EcoS_ESCO41 | Avian cecal content | 50,800 | 46.1 | 1 | 80 | KY619305 |
| vB_EcoM_ESCO47 | Horse dung | 162,477 | 37.9 | 3 | 251 | OM386662 |
| vB_EcoM_ESCO58 | Water | 167,844 | 35.3 | 11 | 282 | OM386666 |

(Fig. 3B). We hypothesized that the recombination event between the genomes of ESCO5 and ESCO37 occurred within these two latter identical regions. No recombinase-coding genes were predicted in ESCO5 and ESCO37. However, both phages encoded an endonuclease (ESCO5_00169, ESCO37_00158) and two helicases (ESCO5_00042, ESCO5_00079, ESCO37_00033, ESCO37_00070), enzymes described in the T4 phage for their role in a homologous recombination pathway (17). REC has all genes predicted in ESCO37 (including 26 genes absent in ESCO5) and has 6 genes specific for ESCO5 that are absent in ESCO37: 3 genes of unknown function (ESCO5_00144, ESCO5_00175, ESCO5_00176), a putative glucose-1-phosphate thymidylyl-transferase gene (ESCO5_00183), a putative dTDP-glucose 4,6-dehydratase gene (ESCO5_00182), and an endo-*N*-acetylneuraminidase gene (ESCO5_00098). This endo-*N*-acetylneuraminidase gene was only present in ESCO5 and REC.

The *Felixounavirus* phages shared 98% identity (>92% coverage; Fig. 3C). In the structure and packaging gene module, differences were observed between the four phages in one of the two tail fiber genes (ESCO45_00034, ESCO49_00001, ESCO50_00074, ESCO56_00001) (average 89.8% coverage, 97.7% identity).

The three *Tevenvirinae* virus genomes, which shared about 50% identity at the nucleotide level, did not exhibit highly conserved synteny (Fig. 3D). However, despite the fact that most of the genes were scattered and some were unrelated, the genome architecture was similar between *Dhakavirus* ESCO47 and *Tequatrovirus* ESCO58 and showed synteny. The most conserved region is the structural module (79% identity, 66% coverage). As demonstrated in Hendrix et al. (18), phages share a common genetic pool through horizontal gene transfer events, which could explain this similarity between ESCO47 and ESCO58.

The two *Tequintavirus* phages shared 84% identity (Fig. 3F). The most striking difference was present in the structural genes: a tail fiber protein-coding gene of ESCO30 (ESCO30_00044) and two tail fiber-coding genes (ESCO40_00133, ESCO40_00131) separated by a gene of unknown function (ESCO40_00132) in ESCO40.

**Phages have narrow to broad host ranges.** The phage host range was determined on a panel of APEC strains belonging to the major serogroups associated with avian colibacillosis (19) (O1, O2, O5, O8, O18, and O78) and on three of the most common *Salmonella* serovars encountered in poultry farming (*S.* Enteritidis, *S.* Typhimurium, and *S.* Infantis) (20) (Fig. 4).

Surprisingly, no phage, not even the *Felixounavirus* phages, had lysed the three *Salmonella* strains tested, even though the *Felixounavirus* type species, FelixO1, isolated

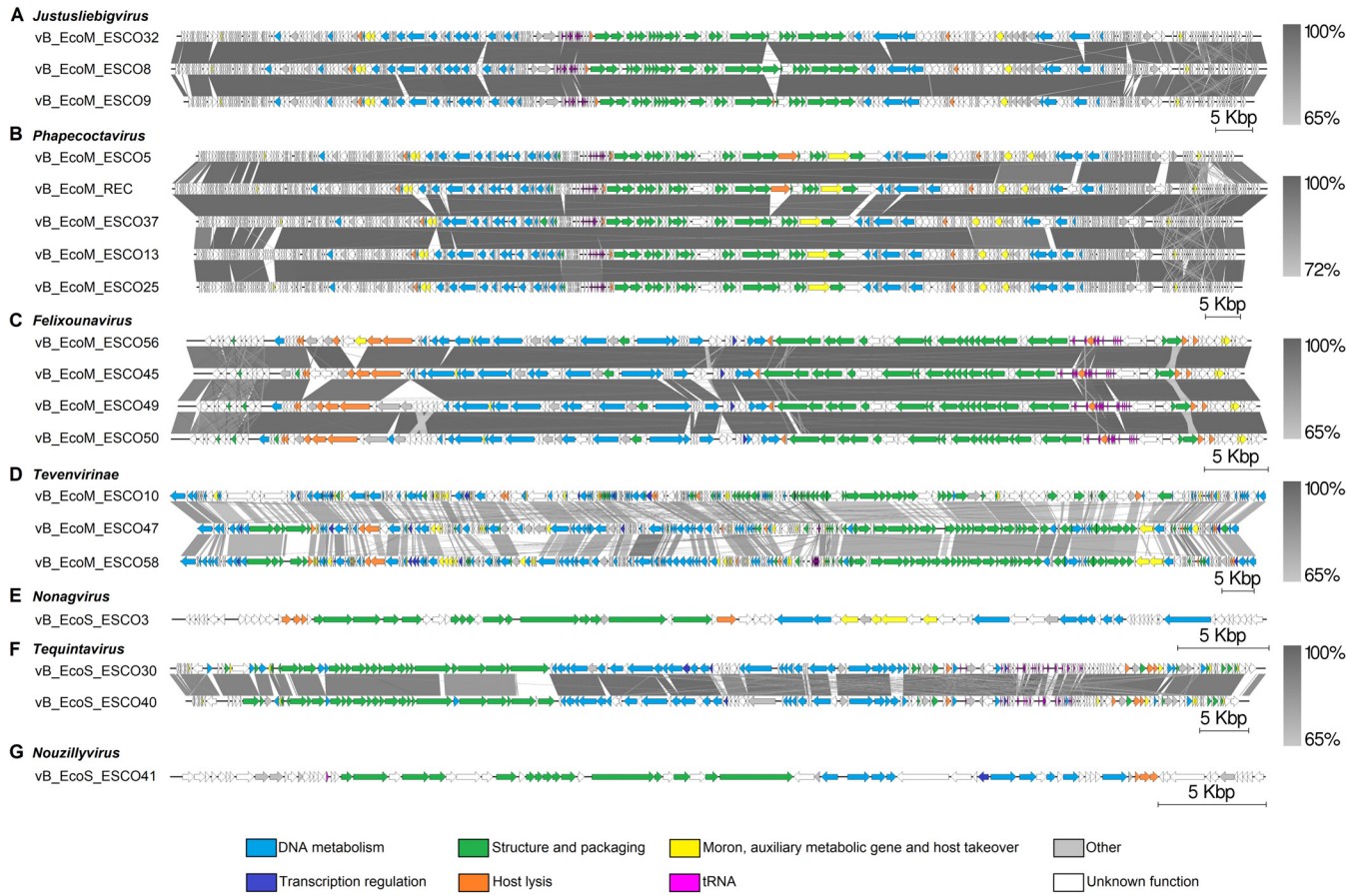

**FIG 3** Genome structures of isolated phages and comparison of each genus or subfamily. The different modules are DNA metabolism (blue); transcription regulation (royal blue); structure and packaging (green); host lysis (orange); moron, auxiliary metabolic gene and host takeover (yellow); other (gray); hypothetical protein and unknown function (white); and tRNA (pink). Arrows represent open reading frames with direction. Level of acid nucleic identity (%) is shown by the gradient scales.

from an *S. paratyphi* strain, is described to have a broad host range (21–23) that includes strains belonging to the serovars tested in the present study. Lipopolysaccharide (LPS) has been described as the bacterial receptor of *Felixounavirus* phages (24). There is great diversity in the host range of *Felixounavirus* phages, which may be associated with modifications in the genomes, for instance, in the baseplate and tail proteins. Indeed, only 50% shared identity was observed between the Felix01 gp77 tail fiber and one of the other tail fibers (ESCO45_00034, ESCO49_00001, ESCO50_00074, and ESCO56_00001). Moreover, the *Felixounavirus* phages described by Maffei et al. (13) did not lyse either of the two *Salmonella* strains tested.

The *Nouzillyvirus*, *Tequintavirus* ESCO30, and two *Felixounavirus* phages (ESCO45 and ESCO56) were able to infect only one strain each (an O78 APEC strain for the *Nouzillyvirus* and a K-12 strain for ESCO30, ESCO45, and ESCO56). Despite the seven tail fibers predicted in ESCO30, unlike other phages of the *Tequintavirus* genus which have three (T5gp085, T5gp133 and pb5/T5gp157) (25), ESCO30 was unable to infect any of the tested APEC strains. In T5 and related phages, the pb5 protein (RBP) binds to the bacterial receptor FhuA (26), which is widely present in *E. coli* strains. Because the Pb5 and T5gp133 genes were absent from *Tequintavirus* ESCO30, it could be hypothesized that ESCO30 did not recognize FhuA as a receptor. T5-like phages recognize other receptors, BtuB and FepA, which are widely distributed within *E. coli* species (13). Because ESCO30 has a narrow host range, this suggests that it targets another receptor that is less distributed or less accessible.

*Dhakavirus*, *Mosigvirus*, *Justusliebigvirus*, two *Felixounavirus* (ESCO49 and ESCO50), and *Tequintavirus* ESCO40 have narrow host ranges, forming plaques on only 3% to 9% of the

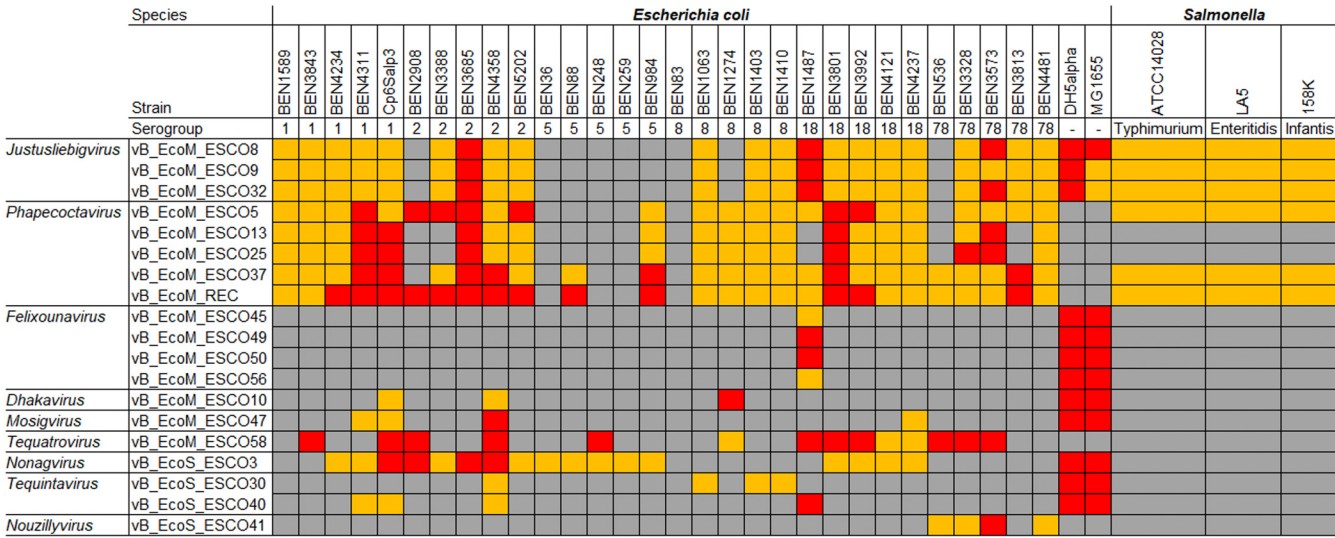

**FIG 4** Phage host range. Host range was determined by a spot test assay against 30 avian-pathogenic *Escherichia coli* (APEC) strains (BEN no. and Cp6Salp3), *E. coli* DH5α and MG1655 laboratory strains, and 3 *Salmonella enterica* subsp. *enterica* strains. Three bacterial effects were observed and are shown in this matrix: plaque lysis formation (red), lysis without plaque (orange), and no lysis (gray).

tested APEC strains. This was a surprising result, especially for the three phages belonging to the genus *Justusliebigvirus*. Indeed, the type phage of this genus, phi92, exhibits a wide host range due to its multivalent adsorption apparatus (27). ESCO8, ESCO9, and ESCO32 have four tail fibers highly similar to those of phi92 (gp141, gp142, gp150, and gp151), with >97% shared identity. However, they did not possess a homolog to the phi92 gp143 gene, which encodes an endo-*N*-acetylneuraminidase (EndoN) located at the distal end of the fibers (28). EndoN would degrade capsules composed of α2.8/α2.9-type sialic acid, such as *E. coli* K92 capsule (28), but possesses a greater ability to bind to the α2.8, thus digesting the *E. coli* K1 capsule. At the gp143 position, ESCO32 had no gene, while ESCO8 had an additional tail spike (ESCO8_00245). ESCO9 had a truncated gene (249 bp) predicted with an endo-*N*-acetylneuraminidase activity, which was probably nonfunctional. Considering the important role of depolymerase enzymes in the phage infection cycle, by cleaving the osidic bonds of polysaccharides (29), their absence could explain the low infection capacity of ESCO8, ESCO9, and ESCO32.

Phages belonging to *Tequatrovirus* (31% of APEC strains lysed) and *Nonagvirus* (11%) and the five *Phapecoctavirus* phages (14% to 37%) had broader host ranges. Like other *Tequatrovirus* phages, ESCO58 has a high infectivity (30, 31). According to Maffei et al. (13), the putative receptors of *Nonagvirus* phage should be similar to those of *Tequintavirus* (*O*-antigen glycan and surface proteins FhuA, LptD, or BtuB). The endosialidase chaperone (ESCO3_00080) encoded by the ESCO3 *Nonagvirus* would allow it to degrade the capsule and reach the receptor, explaining its higher infectivity. As for *Justusliebigvirus* phages, the receptors of *Phapecoctaviruses* could putatively be the enterobacterial common antigen (ECA) and the first glucose of the outer LPS core (13). ESCO5 and REC carried a depolymer-ase (ESCO5_00098, REC_00244) that was 96% identical to endoN of phi92. Since it allows the degradation of K1 and K92 capsules, the presence of this enzyme could explain the wide host range of these phages. Phage polysaccharidases are increasingly being studied as therapeutic agents. Indeed, removal of the capsule sensitizes the bacteria to the host's immune system and may cause it to lose its pathogenicity. Purified depolymerases have already proven to be effective against systemic infections by improving the survival rate of mice (40% to 70%) (32, 33). Furthermore, no resistance against a phage depolymerase has been observed (34). However, their drawback remains their very high specificity, often restricted to few isolates. Moreover, *Phapecoctavirus* phages, which were found by metage-nomic analyses in a Microgen ColiProteus cocktail used in human medicine (35), could have high therapeutic potential.

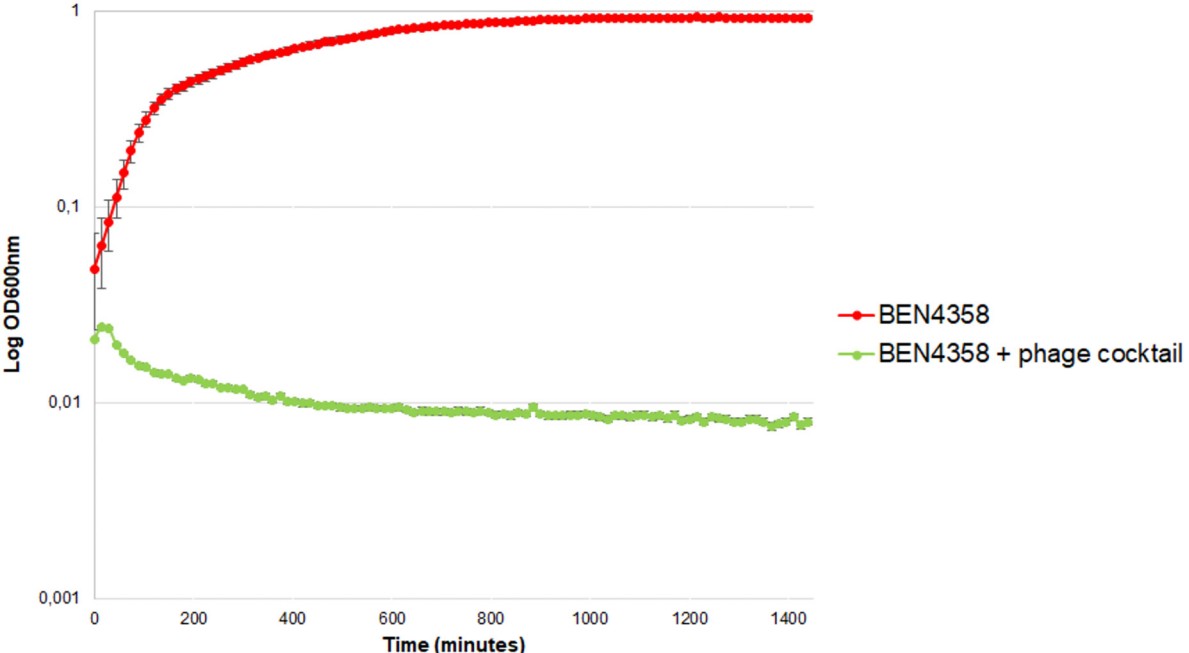

**FIG 5** *In vitro* lytic activity of phage cocktail against APEC BEN4358 strain. An overnight bacterial culture was diluted ($2 \times 10^6$ CFU/mL) and infected with phage cocktail (ESCO9, REC, ESCO3, ESCO10, ESCO30, ESCO41, ESCO47, and ESCO58) at a multiplicity of infection (MOI) of 10 (green curve). Non-phage-treated bacterial cultures were used as a positive control (red curve). Bacterial growth was recorded by monitoring the optical density at 600 nm ($OD_{600}$) every 10 min for 24 h. Each point represents the mean from triplicate experiments, and standard deviations ($\pm$SD) are shown by vertical lines.

**The eight-phage cocktail fully inhibits growth of an *E. coli* O2 strain and reduces chicken embryo mortality by 90%.** To demonstrate the therapeutic potential of isolated phages, a phage combination was tested *in vitro* and in a chicken embryo lethality assay (CELA). To account for the phage diversity within the collection, a cocktail was designed containing eight phages belonging to eight different genera (genus *Felixounavirus* was not included because it has the narrowest host range). BEN3685 proved to be the most sensitive to phages (nine phages from three different genera), followed by BEN4358 (five phages from four genera). However, in CELA, BEN3685 was shown to be less virulent (50% embryo mortality at 5 days postinfection [dpi]) than BEN4358 (70% mortality at 5 dpi; data not shown). Therefore, the latter strain was chosen for the *in ovo* experiment. Four phages of the cocktail replicated on this strain by plaque lysis formation (REC, ESCO3, ESCO47, and ESCO58), three phages induced lysis without replication (ESCO9, ESCO10, and ESCO30), and ESCO41 was not active on the strain.

The *in vitro* lytic activity of the phage cocktail against the BEN4358 strain was monitored by measuring the optical density at 600 nm ($OD_{600}$) over 24 h of incubation. At an MOI of 10, the phage cocktail fully inhibited the growth of BEN4358 in LB medium (Fig. 5).

Next, the efficacy of this phage cocktail was evaluated *in vivo* (Fig. 6). CELA allows us to examine the virulence of the bacterial strain and evaluate the efficacy of phage therapy (36), and it is relevant because APEC strains cause embryo mortality by contaminating the vitelline membrane (37). The phage cocktail treatment allowed 90% of the chicken embryos to survive a 6-day infection by BEN4358, in contrast with the non-phage-treated bacterial control group in which all of the embryos had died by day 4. The addition of ceftiofur allowed 100% of the chicken embryos to survive. When only the Dulbecco's phosphate-buffered saline (DPBS) or phage cocktail was inoculated, no embryos died, showing the innocuity of the *in ovo* inoculation and the phage cocktail. However, two embryos died on day 1 after the injection of the ceftiofur alone. Thus, in this study, we showed that the cocktail was effective in treating avian colibacillosis in chicken embryos. In further studies, the efficacy of the phage cocktail at treating older animals will be assessed.

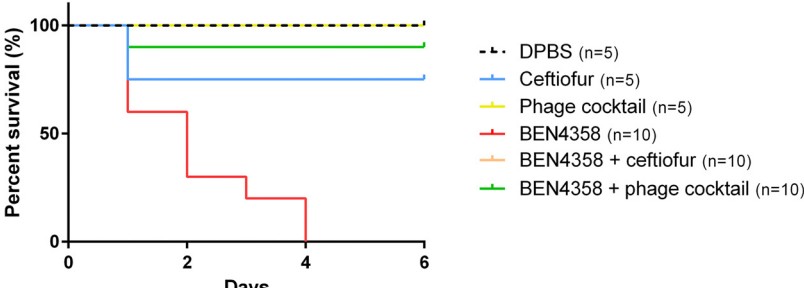

**FIG 6** *In ovo* therapeutic efficacy of phage cocktail against APEC BEN4358 strain. Kaplan-Meier survival curves of embryonated chicken eggs. Some eggs were inoculated with 266 CFU of *E. coli* BEN4358 strain alone (red curve), BEN4358 with 6,000 PFU of the phage cocktail (ESCO9, REC, ESCO3, ESCO10, ESCO30, ESCO41, ESCO47, and ESCO58) added (green curve), or 1 mg/mL ceftiofur (orange curve) 2 h after the bacterial inoculation. Inoculation of only Dulbecco's phosphate-buffered saline (black curve), 1 mg/mL ceftiofur (blue curve), or 6,000 PFU of the phage cocktail (yellow curve) were used as controls.

**Conclusion.** Phage therapy is a promising alternative to antibiotics to control avian colibacillosis, particularly because of the specificity of phages against bacterial strains or species and thus the absence of secondary effects on the host microbiota. This study reports the isolation and characterization of 19 phages against APEC. The absence of virulence genes (toxins or antibiotic resistance genes) and lysogenic cycle-associated genes in these phages indicates that they are probably lytic and can be used in phage therapy. Genome and host-range analyses showed that there were intra-genus variations among the closely related phages. In addition, some phages encoded polysaccharidases, and these were found to be among the most lytic against APEC strains, demonstrating the importance of phage characterization. Lastly, a cocktail of eight phages was shown to be a promising candidate for biocontrol of avian colibacillosis, as it was found to be highly effective at killing a pathogenic APEC serotype.

## MATERIALS AND METHODS

**Bacterial strains and culture conditions.** Thirty-nine *E. coli* and three *Salmonella* strains were used in this work (Table S2). *E. coli* strains with a BEN number (own collection) and strain Cp6Salp3 (38) are APEC strains collected in the field, and DH5$\alpha$ and MG1655 are laboratory strains. *Salmonella enterica* subsp. *enterica* LA5 (39), ATCC 14028 (40), and 158K (41) are virulent and widely used as type strains. All strains were grown in lysogeny broth (LB, Miller formula) medium at 37°C with shaking at 180 rpm and stored at −80°C in 15% (vol/vol) glycerol.

**Phage isolation.** Various environmental samples (sewage, pond water, river water, avian cecal content, horse dung) were collected in France between 2013 and 2017. Samples were homogenized and centrifuged at 8,000 × $g$ for 10 min at 4°C to remove impurities, followed by supernatant filtration through 0.45- and 0.2-$\mu$m pore microfilters (ClearLine) to remove bacterial debris. The presence of phages was determined by a time-kill curve method (42) using a Microbiology Reader Bioscreen C (Thermo Fisher Scientific) in 100-well honeycomb, sterile covered microplates by OD$_{600}$ monitoring. Each sample was tested on 16 bacterial strains: DH5$\alpha$, which carries no prophage and possesses no restriction/modification system, and a panel of APEC strains of the main serogroups responsible for colibacillosis (19) (Table S2). Overnight bacterial cultures were diluted 1:50 in LB medium and incubated at 37°C with shaking (180 rpm) until reaching the logarithmic growth phase (OD$_{600}$ ∼ 0.4). Next, 150 $\mu$L of 1:20 diluted bacterial cultures, 10 $\mu$L of filtered environmental sample, and 140 $\mu$L of LB medium with MgSO$_4$ (10 mM) and CaCl$_2$ (1 mM) was distributed in a 100-well honeycomb plate. LB medium without bacteria and bacterial cultures without environmental samples were included as controls. The OD$_{600}$ was automatically recorded every 10 min using a 600-nm filter over 24 h, with soft shaking before the measurements.

The observation of bacterial growth retardation revealed the presence of phages in the environmental samples. To amplify phages from those samples, 500 $\mu$L of filtered environmental sample was added to 200 $\mu$L bacterial culture (OD$_{600}$ ∼ 0.4) in the presence of MgSO$_4$ (10 mM) and CaCl$_2$ (1 mM) and incubated overnight at 37°C with shaking (180 rpm). The culture was centrifuged at 3,000 × $g$ for 15 min at room temperature and the supernatant was filtered using a 0.2-$\mu$m pore microfilters to remove bacterial cells. The presence of phages was detected by spot test assays based on methods described by Kutter et al. (43), with some modifications: 200 $\mu$L of *E. coli* strain (OD$_{600}$ ∼ 0.4) was added to 5 mL LB agarose 0.5% (wt/vol) maintained at 55°C, CaCl$_2$ (1 mM), MgSO$_4$ (10 mM), and 30 $\mu$M 2,3,5-triphenyltetrazolium chloride (TTZ), and poured onto a 1.5% (wt/vol) LB agar plate. Then, 10 $\mu$L of 10-fold dilutions of the lysates in TM buffer (Tris-HCl 10 mM, MgCl$_2$ 10 mM [pH 8]) were spotted onto the solidified bacterial lawn and incubated

overnight at 37°C. Isolated plaques were picked and amplified using the same method as described above. Lysates were stored at 4°C.

**Phage purification.** According to a previously described method (44), 500 mL of phage lysate was precipitated with 10% (wt/vol) PEG 6000 (Sigma-Aldrich) and 0.5 M NaCl and incubated overnight at 4°C with agitation. Bacterial cells were removed by centrifugation at 4,400 $\times$ $g$ (Beckman Coulter, Aventi JE Centrifuge) for 30 min at 4°C. The pellet was resuspended in TM buffer. PEG was removed twice by adding an equal volume of chloroform, mixed by gentle inversion for 30 s, and centrifuged at 3,800 $\times$ $g$ (Beckman, GS-15R Centrifuge) for 10 min. Then, 0.75 g of cesium chloride (CsCl) was added per mL of the aqueous phase containing phage particles (45). The solution was centrifuged at 31,000 rpm (Beckman Coulter, Optima L80 XP Ultracentrifuge) for 48 h at 10°C in a SW 41 Ti rotor. The phage band was recovered by piercing through the ultracentrifuge tube with a sterile 25-G needle (Terumo Corporation). To remove CsCl, the sample was transferred in a dialysis cassette G$_2$ (Thermo Fisher Scientific) and dialyzed against 1.5 L of TM buffer at 4°C overnight.

**Transmission electron microscopic analysis.** The CsCl-purified phage suspension was placed onto a Formvar/carbon-coated TEM grid for 10 min in a moist chamber. The grid was then washed three times by touching the grid surface with drops of deionized water. Remaining water was wicked away by touching filter paper to the side of the grid. A drop of 1% uranyl acetate was applied to the grid for 1 min. The grid was air-dried at room temperature and stored for subsequent TEM imaging. All TEM grids were evaluated on a JEOL JEM-1011 (100 kV). All digital images were acquired using an CMOS GATAN Rio 9 camera system by the Infrastructures en Biologie Santé et Agronomie (IBiSA) electronic microcopy platform (Tours, France).

**Phage DNA extraction.** Genomic DNA was extracted using the phenol-chloroform method: 28 $\mu$L of 0.5 M EDTA, 35 $\mu$L of 10% SDS, and 2 $\mu$L of proteinase K (25 mg/mL) (Thermo Fisher Scientific) were mixed with 600 $\mu$L of purified phage solution and incubated at 65°C for 1 h. Next, 133 $\mu$L of 3 M KCl was added, and the solution was centrifuged at 16,100 $\times$ $g$ (Centrifuge 5415R; Eppendorf) for 10 min at room temperature. An equal volume of phenol-chloroform was added to the supernatant, which was mixed by gentle inversion for 30 s and centrifuged at 16,100 $\times$ $g$ for 13 min at room temperature. The aqueous phase containing phage DNA was collected, mixed with 2.5 volumes of absolute ethanol, stored at –20°C, and incubated for 30 min at 4°C. After centrifugation at 16,100 $\times$ $g$ for 15 min at room temperature, the pellet was washed with 70% ethanol. Lastly, the pellet was air-dried at room temperature, resuspended in 50 $\mu$L water, and stored at –20°C for use.

**Genome analysis and sequencing.** Phage genomes were cut with the restriction enzymes HincII (BioLabs), and digested fragments run on a 0.8% agarose gel in 0.5$\times$ TBE buffer (Tris 44 mM, borate 44 mM, EDTA 1 mM [pH 8.3]). Fingerprints were analyzed using BioNumerics software (Applied-Maths) and dendrograms were generated using the Dice coefficient and Unweighted Pair-Group Method with Arithmetic Average (UPGMA).

Next, we performed genomic DNA extraction of phages from a high-titer phage lysate. DNA phage libraries were prepared with the Nextera kit (Illumina) and sequenced to 2 $\times$ 250-bp read length on a MiSeq system (Illumina) by the DNA Sequencing Facility at the University of Cambridge (United Kingdom).

**Bioinformatics analysis.** The raw sequence reads in FastQ files were quality-filtered using Sickle (46). Next, reads were assembled with SPAdes (v3.5.1) (47) and phage genomes were resolved as a single contig. Coding sequences and tRNA genes were predicted and automatically annotated using Prokka (48). A manual annotation was performed using VIRFam to predict head-neck-tail module genes (49), and the BLASTp (50) algorithm with identity percentage of >60% was used to predict the best gene annotation with protein sequences submitted to the NCBI database.

The phage taxonomy was determined by using BLASTn (51) against the NCBI database, and VIRIDIC (Virus Intergenomic Distance Calculator) (52) was used to score the phylogenomic distances. Phage genomes in the database with an identity and coverage percentage of >95% were designated the same species. A phylogenetic tree based on genome sequence similarities computed by tBLASTx was constructed using VIPtree (v2.0) (53). Genomic structures and comparison maps of phages belonging to the same genus were made using EasyFig (v2.2.3) (54). The open reading frame categories were based on those of PHROG (55).

**Primer design and phage PCR.** To ensure the purity of the phage lysates, we designed primers specific to each phage. First, phage genomes were aligned by Mauve (56) using the bioinformatics software platform Geneious, and primers were determined *in silico* in genes specific for each phage. The primers used in this work are described in Table 3. For some phages, two pairs of primers were used. Indeed, for the REC phage, PCA240/PCA241 and PCA248/Esco37_0008R were used.

PCRs were performed as follows: each mix of 25 $\mu$L final volume contained 2 $\mu$L lysate, 5 $\mu$L of 5$\times$ Green GoTaq reaction buffer (Promega), 1.5 $\mu$L MgCl$_2$ (25 mM) (Promega), 0.5 $\mu$L dNTP (10 $\mu$M) (Promega), 1.25 $\mu$L of each primer (10 $\mu$M) (Eurogentec) and 0.2 $\mu$L GoTaq DNA polymerase (Promega). Extracted phage DNAs were used as positive controls for the respective PCRs. The thermocycling conditions were an initial denaturation of the DNA template for 4 min at 94°C, followed by 30 cycles of denaturation for 30 s at 94°C, annealing at the melting temperature of the primer for 30 s, and extension at 72°C (30 s per 500 bp product length), and a final extension for 5 min at 72°C (ProFlex, Applied Biosystems). PCR products were separated by electrophoresis in 1% agarose (Eurogentec) gels in 0.5$\times$ Tris-acetic acid-EDTA (TAE) buffer (Sigma-Aldrich) with 1/500 Midori Green Advance (Dutcher), and visualized under UV using GelDoc Go Imaging System (Bio-Rad).

When a lysate contained more than one phage, a spot test assay was performed to obtain isolated

**TABLE 3** Primers designed and used in this study

| Phage | Primers | Sequence (5′→3′) | Product size (bp) | Annealing temp (°C) |
|---|---|---|---|---|
| ESCO8 | PCA286 | TATGGCTGCCGAACACCATT | 787 | 54 |
| | PCA287 | TGCGTAGTGAATGCCCCGTT | | |
| ESCO9 | PCA288 | CGATCCGCTGTTTCCCTGAT | 343 | 53 |
| | PCA289 | CCGTGTGTGATGCTTGGTTC | | |
| ESCO32 | PCA290 | TTGTGGTAGTGGGCGATGAA | 1134 | 53 |
| | PCA291 | ATGGAAGCTCTGCCCCATAC | | |
| ESCO5 | PCA240 | CGAACCAACACCACATGCAG | 1140 | 53 |
| | PCA241 | CGAACCAACACCACATGCAG | | |
| | PCA296 | CTGTGCACGGTGTATGGGAT | 412 | 53 |
| | PCA297 | CTCCCAGCTTTTTCCCCTGT | | |
| ESCO13 | PCA242 | AGTGGAAGTGTATTGCGGGG | 166 | 53 |
| | PCA243 | GGTGCCGGACTTCTTCTCAA | | |
| ESCO25 | PCA244 | CACCGGACGACATTGAAACG | 130 | 54 |
| | PCA245 | GCAAGATGACCATAAGCCGC | | |
| | PCA246 | GCTTGTGCATGGGTAGGGAT | 74 | 53 |
| | PCA247 | TTGCTCTTTGTTGCGCGTT | | |
| ESCO37 | PCA248 | TTGCCAAGCGATTGGAAACG | 264 | 53 |
| | Esco37_0008R | GCTTTTTCAAAACTAGAACA | | |
| ESCO45 | PCA262 | GGTGGTTGAGCATAGCCTGT | 715 | 52 |
| | PCA263 | AGGCATGGTATGGTTGGGTG | | |
| ESCO49 | PCA274 | CCCTTTCGGGTTATGGGCTT | 486 | 53 |
| | PCA275 | AAAATGGCAACTATCCGCGC | | |
| ESCO50 | PCA276 | ACCATTTGTCGCTACGTGGT | 1039 | 53 |
| | PCA277 | TAGTAGCCCCAGATGCAGGT | | |
| ESCO56 | PCA278 | CCTTCACAGTGCCAGCCTTA | 325 | 53 |
| | PCA279 | AGGGAGTACACCTGCATTGC | | |
| ESCO3 | PCA264 | CGTTTGCAAGCTATCTCGGC | 729 | 54 |
| | PCA265 | CGGACGTTCGTTTGTTGCTT | | |
| ESCO10 | PCA266 | TGACGTAGGCGCCAATTTCT | 607 | 54 |
| | PCA267 | TCGGTCGTTAAACAGGCGAA | | |
| ESCO30 | PCA258 | ACCGCCCTATTACTGACCCT | 898 | 53 |
| | PCA259 | GCATCTGTAGAGTCCGCTCC | | |
| ESCO40 | PCA268 | CAGCCAAAACTGGTGAACCG | 641 | 54 |
| | PCA269 | GCTGTGGTCTGGATGCTCTT | | |
| ESCO41 | PCA270 | CTGATGCTTGGGGATGGGTT | 980 | 56 |
| | PCA271 | CGTTGCGCTATCTCCGTAGT | | |
| ESCO47 | PCA272 | TGGCTGACGACGTTTTAGCT | 802 | 54 |
| | PCA273 | GCTTCGTCCAACTGGTCTGA | | |
| ESCO58 | PCA280 | TCTGCGAGCACAACATACGT | 672 | 53 |
| | PCA281 | TTTCACCGCAGTTTCAGGGT | | |

plaques. Each plaque was collected with a toothpick and mixed in 15 $\mu$L of sterile water. Two $\mu$L of this mix were tested by PCR. A plaque positive for the phage of interest was amplified to obtain a new lysate.

**Host range determination.** The host range was determined using a spot test assay, as described above, on 35 different bacterial strains (Table S2). Briefly, 10 $\mu$L of each phage lysate ($\sim$10$^9$ PFU/mL) and 10-fold serially diluted in Dulbecco's phosphate-buffered saline (DPBS; Sigma-Aldrich) before being spotted onto bacterial lawns and incubated overnight at 37°C. Phage infection was determined by visual examination of the plates for plaques. The turbidity plaques were also assessed and graded (clear, opaque, lysis without plaques, or no lysis).

**Lytic activity of phage combinations.** A BEN4358 or Cp6Salp3 overnight culture was diluted 1:50 in LB medium and incubated at 37°C with shaking (180 rpm) until reaching the logarithmic growth phase (OD$_{600}$ $\sim$ 0.4). Wells of a 100-well honeycomb plate were filled with 20 $\mu$L bacterial culture adjusted to 2 $\times$ 10$^6$ CFU/mL, 20 $\mu$L of single phage lysate or phage cocktail (2 $\times$ 10$^7$ PFU/mL), and 140 $\mu$L of LB medium MgSO$_4$ (10 mM) and CaCl$_2$ (1 mM). The plate was incubated in a Microbiology Reader Bioscreen C, at 37°C with shaking. The OD$_{600}$ of each sample was monitored at 10-min intervals for 24 h. Data represent three independent experiments.

**Chicken embryo lethality assay.** To determine the efficacy of phage cocktail against an APEC strain, we performed a chicken embryo lethality assay based on a method described by Trotereau and Schouler (36), with some modifications. Briefly, specific-pathogen-free embryonated White Leghorn eggs were incubated in an egg incubator (MG140/200, FIEM), at 37.8°C and 45% humidity. On day 11, eggs (provided by the Infectiology of Farm, Model and Wildlife Animals Facility (PFIE) doi.org/10.15454/1.5572352821559333E12; member of the National Infrastructure EMERG'IN) were candled to eliminate infertile eggs and eggs with dead embryos. On day 12, a BEN4358 overnight culture was adjusted in DPBS to 2 $\times$ 10$^3$ CFU/mL. The eggshell was

washed with 70% ethanol and pierced by a sterile 18-G needle, without piercing the shell membrane, about 2 mm from the end of the air sac. Using a sterile 25-G needle and syringe, 100 $\mu$L of bacterial inoculum was injected into the allantoic cavity. The holes were sealed with a small round sticker, then the eggs were incubated in the egg incubator, with the air space pointing up. Two hours after the bacterial inoculation, eggs were inoculated with 100 $\mu$L of phage suspension adjusted to $4 \times 10^4$ PFU/mL in DPBS (10 eggs) or 100 $\mu$L of 1 mg/mL ceftiofur (Sigma-Aldrich) (10 eggs). As controls, 5 eggs were inoculated with only 100 $\mu$L DPBS, 100 $\mu$L 1 mg/mL ceftiofur, or 100 $\mu$L phage cocktail. The eggs were candled daily for 6 days to monitor embryo mortality. Data are presented by Kaplan-Meier curves and analyzed by a log-rank test.

**Ethics statement.** The animal experimental plan was evaluated and approved by the local ethic committee (CEAA Val de Loire) and the French Ministry for Higher Education and Research under APAFIS no. 26717-2020072413185773 v4.

**Data availability.** The genome sequences of coliphages (ESCO3, ESCO8, ESCO9, ESCO10, ESCO25, ESCO30, ESCO32, ESCO37, ESCO40, ESCO45, ESCO47, ESCO49, ESCO50, ESCO56, ESCO58, REC, ESCO5, ESCO13, and ESCO41) are available in GenBank under the following accession numbers: OM386652, OM386653, OM386654, OM386655, OM386656, OM386657, OM386658, OM386659, OM386660, OM386661, OM386662, OM386663, OM386664, OM386665, OM386666, OM386667, KX664695, KX552041, and KY619305, respectively.

## SUPPLEMENTAL MATERIAL

Supplemental material is available online only.
**SUPPLEMENTAL FILE 1**, PDF file, 0.5 MB.

## ACKNOWLEDGMENTS

This study has received funding from ANR through the ANIHWA (Animal Health and Welfare: ERA-Net), ANTIBIOPHAGE project ANR-14-ANWA-0003-03 and from DGAL through the EcoAntibio2 COLIPHAVI project. M.N. was supported by a Ph.D. stipend from Tours University. A.T. was supported by a training grant from the Fédération de Recherche en Infectiologie (FéRI).

C.S. contributed to conceptualization, funding acquisition, methodology, project administration, supervision, validation, and review and editing of the manuscript. M.N. contributed to formal analysis, investigation, methodology, validation, visualization, data curation, writing of the original draft, and review and editing of the manuscript. A.T. contributed to investigation, visualization, and writing of the original draft. A.C. contributed to formal analysis, investigation, validation, and visualization. A.M., P.V., J.W., R.L., and R.A. contributed to review and editing of the manuscript.

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
