## [Reviewer comments · Microbiology Spectrum]

Microbiology Spectrum

Isolation and characterization of a novel phage collection against avian pathogenic *Escherichia coli*

Marianne Nicolas, Angéline Trotereau, Antoine Culot, Arshnee Moodley, Robert Atterbury, Jeroen Wagemans, Rob Lavigne, Philippe Velge, and Catherine Schouler

Corresponding Author(s): Catherine Schouler, Institut National de Recherche pour l'Agriculture l'Alimentation et l'Environnement

Review Timeline:

Submission Date:	October 24, 2022
Editorial Decision:	December 12, 2022
Revision Received:	February 9, 2023
Editorial Decision:	April 6, 2023
Revision Received:	April 11, 2023
Accepted:	April 12, 2023

Editor: Thomas Denes

Reviewer(s): The reviewers have opted to remain anonymous.

Transaction Report:

DOI: <https://doi.org/10.1128/spectrum.04296-22>

December 12, 2022

Dr. Catherine Schouler
Institut National de Recherche pour l'Agriculture l'Alimentation et l'Environnement
Nouzilly 37380
France

Re: Spectrum04296-22 (Isolation and characterization of a novel phage collection against avian pathogenic Escherichia coli)

Dear Dr. Catherine Schouler:

Thank you for submitting your manuscript to Microbiology Spectrum. Your manuscript was reviewed by one expert in the field, who has provided extensive constructive feedback. I have also closely reviewed the manuscript and believe that thoroughly addressing the comments of the reviewer will greatly improve the manuscript. A majority of what I picked up on in my read-through was covered by the reviewer; however, I do have a few comments to add:

1. Line 182: please revise "probably strictly lytic" as defining a phage as strictly lytic requires strong evidence
2. I could not find the legend for Figure 2. It was also unclear how such a high level comparative genomics analysis fits into this study. The section starting lines 154 seemed to be addressing questions outside the scope of this work (as I understood it). In my opinion, it would be more in-scope to have both the references for the phage genera compared alongside the phages from this study; this could be a more focused analysis including only the relevant genera (separate or combined).
3. Figure and Table callouts should be consistent throughout manuscript.

Link Not Available

Sincerely,

Thomas Denes

Journals Department
Reviewer comments:

Reviewer #1 (Comments for the Author):

Summary

In this work, the authors isolated and genetically characterise phages infecting avian pathogenic E. coli (APEC) and use them to for a cocktail to kill this pathogenic bacterium in an in ovo model, demonstrating increased survival of the embryos.

Overall impression and shortcomings

While the phage isolation and host range analysis are well performed, the bioinformatic analysis is rudimentary and lacks argumentation for choosing lysins and spannin for analysis. In addition, what is the rationale behind using 8 phages for a cocktail? In addition, there is no need to repeat data that are shown in the tables, only highlight finding that are unusual or new. This manuscript can be shortened substantially to avoid repeating information and increase clarity.

Specific minor and major comments

Abstract and importance

Line 26: Specify the method behind the "genome taxonomy studies"

Line 27-28: It is not clear why these functions are investigated, please explain.

Line 28: the word "encoding" is not appropriate here, use carry or another word for the enzymatic activity. Also, not clear why this is important to mention in the abstract?

Line 29: Usually serotypes are used rather than serogroups. Does "consisting of 26 of the 30 APEC strains" means that 4 strains were resistant to all phages? I think "consisting of" is the confusing word here as it refers to the word "tested" and not the strains.

Line 30: Again, please be more specific, what kind of enzymes are we talking about? Are these lysins or are the enzymatic activity part of a receptor binding protein?

Line 30-31: It's a bit weird mentioning the REC phage here, I would suggest that you explain this before going to the host range, so move to line 29.

Line 52: Add "to survive the APEC infection". Delete "we think that this"

Introduction

Lines 72 to 76: It would be relevant to explain more about what are these phages? Taxonomy is changing so if the genomes are out there, please use the current taxonomy and explain more about these phages. Are they o-antigen specific or have other receptors?

Lines 84-86: It is not important that the phages are morphological distinct, more important is their host range and receptors as well as ability to overcome anti-phage defense systems of the host.

Lines 88-92: Again, relevant to add information about what are these phages? It is not particular relevant if they are myoviruses, but the about mentioned biological features are important as well as genera if the sequences are known.

Lines 93: Here you could state more strongly your aim.

Line 95: All phages were new, but one phage did only show 78% identity to a phage in the database, and therefore you can define a new genus. Please revise.

Line 97: The phages do not contain polysaccharides, please clarify if this is an enzymatic activity as part of the RBP?

Results and discussion

Please combine the three first paragraphs into one that focus on the new findings and describes the collection in a short and precise way.

Line 105-106: What is this used for? It seems irrelevant here. Please delete.

Line 108: Please add the RFLP data as supplementary.

Line 109-113: It is weird to introduce the REC phage before the two ancestors. Please move to later.

Line 115-127: this data can be added into a table and shown either as supplementary or as part of one of the other tables. This information is nice, but not something to comment on, unless you see something new to say.

Lines 124-127: This is not important and refer to old taxonomy, please delete this.

Lines 129-131: This can be seen from the tables and can be deleted.

Line 132-134: this could be you starting point

Line 134-138: Delete as part of the table.

Line 154-166: This is relevant and could be moved to line 134, where you refer to figure S2. And then follow by the parts in line 140-146.

The paragraph starting on line 167 concerning the genomes could be shorten and focused on the synteny as the title says. In this paragraph you also present differences in gene content of the phages in each genus. But the question is why is this relevant for your study? Please clarify.

Line 168-182: I would suggest deleting this, as it is not important for the aim and nothing new is found.

Line 191- 205: How can you demonstrate the recombination spot with data and in a figure? Line 206-209: Move to before explaining the REC phage, so you introduce the parental phages first.

In the next paragraph starting on line 227, I miss the rationale for focusing on lysins and spannins? Why are they specifically analyzed? I don't see the relevance for this work. In any case, all genes discussed and analyzed should be mentioned by name, and these should appear in the table as well.

Line 249-258: not relevant and can be deleted.

I think the manuscript would benefit from combining analysis of the adsorption genes with the host range part.

Line 264 and onwards: It would be nice to mention the genus of the phages while presenting and discussing their host range as this can direct you to the potential receptors of the phages. This could also help you relating to relevant literature which I think could provide a more thorough discussion.

Line 269: Tequintavirus do not have many tail fibers, they have three lateral tail fibers and a central spike, recognizing the O-antigen and an outer membrane protein as receptor, respectively. Please rephrase.

Line 272-274: It is not that relevant what others found in terms of numbers of strains infected, as these collections are so different from each other. I find it more relevant to discuss the genus/RBPs/host range.

Line 275-278: Please relate to genus and putative RBPs.

Line 279-280: I respectfully disagree. For therapy, only infection is important and the following parts are not relevant. Lysis without infection is a dead end for the phages, and no propagation will take place, thus omitting the positive effect of auto-replication of the antimicrobial activity. Also, lysis without infection as an antibacterial strategy will require a high level of phages applied and this is simply not feasible.

Line 285-287: What elements? Yes, the depolymerases do have an important role in infection, please elaborate on their function as RBPs.

I like the in ovo experiment, but still I'm not convinced that including phages that do not infect the pathogenic APEC is a good idea. Did you isolate any phages after the experiment? And if yes, what phages were present?

Staff Comments:

Preparing Revision Guidelines

Please return the manuscript within 60 days; if you cannot complete the modification within this time period, please contact me. If you do not wish to modify the manuscript and prefer to submit it to another journal, please notify me of your decision immediately so that the manuscript may be formally withdrawn from consideration by Microbiology Spectrum.

Reply to editor's comments:

1. Line 182: please revise "probably strictly lytic" as defining a phage as strictly lytic requires strong evidence.

- **The sentence was revised as follows: As no integrase nor excisionase-encoding genes were found in the phage genomes, we hypothesized that the 19 phages have a virulent lifestyle (line 155, clean file).**

2. I could not find the legend for Figure 2. It was also unclear how such a high level comparative genomics analysis fits into this study. The section starting lines 154 seemed to be addressing questions outside the scope of this work (as I understood it). In my opinion, it would be more in-scope to have both the references for the phage genera compared alongside the phages from this study; this could be a more focused analysis including only the relevant genera (separate or combined).

- **We thank you for this very helpful suggestion that helps to clarify the message by being more focused. Accordingly, Figure 2 has been revised. The phylogenetic tree was rebuilt with the phages belonging to the 9 genera cited in our study. Legend of the figure has been added. Part of the manuscript referring to this figure has been modified (lines 125 to 132, clean file).**

3. Figure and Table callouts should be consistent throughout manuscript.

- **The callouts of the Figures and Tables were carefully checked and corrected according to their order of appearance.**

Additional correction: The phage taxonomy has evolved again. As the three phi92-like phages (ESCO8, ESCO9, ESCO32) are now classified in the subfamily *Stephanstirmvirinae*, genus *Justusliebigvirus*, we have modified the text (lines 116, 127, 130, 160, 223, 226, 245) and figures (Table 1, Figures 2, 3 and 4) accordingly.

Reply to reviewer's comments:

In this work, the authors isolated and genetically characterise phages infecting avian pathogenic *E. coli* (APEC) and use them to form a cocktail to kill this pathogenic bacterium in an in ovo model, demonstrating increased survival of the embryos.

Overall impression and shortcomings

While the phage isolation and host range analysis are well performed, the bioinformatic analysis is rudimentary and lacks argumentation for choosing lysins and spannin for analysis. In addition, what is the rationale behind using 8 phages for a cocktail? In addition, there is no need to repeat data that are shown in the tables, only highlight findings that are unusual or new. This manuscript can be shortened substantially to avoid repeating information and increase clarity.

We thank you for your constructive and helpful comments. We agreed that presenting lysins and spannins in this study is not relevant. We then focused on enzymes that could play a role in the first steps of the phage lifecycle, in line with hypothesis about the

different ability of the phages to infect few or various avian pathogenic *E. coli* strains. According to your comments, the manuscript has been reorganized and shortened.

Specific minor and major comments

Abstract and importance

Line 26: Specify the method behind the "genome taxonomy studies"

- **We corrected the sentence (line 26, clean file). The affiliation of a phage to a genus was done on the basis of sequence homologies.**

Line 27-28: It is not clear why these functions are investigated, please explain.

- **The section related to enzymes involved in bacterial lysis at late steps of phage lifecycle was deleted. We now only referred to phage associated-enzymes such as tail-associated depolymerases that could allow the phages to reach more easily their(s) bacterial receptor(s). The presence or absence of such enzymes may partly explained the observed differences in phage infectivity**

Line 28: the word "encoding" is not appropriate here, use carry or another word for the enzymatic activity. Also, not clear why this is important to mention in the abstract?

- **As suggested, the word carrying was used (line 31). We mentioned this in the abstract because, as indicated above, presence or absence of tail-associated depolymerases may partly explained the ability of phages to infect various strains.**

Line 29: Usually serotypes are used rather than serogroups. Does "consisting of 26 of the 30 APEC strains" means that 4 strains were resistant to all phages? I think "consisting of" is the confusing word here as it refers to the word "tested" and not the strains.

- **The term serotype referring to the combination of O, H and K typing, we use the term serogroup because we only mentioned the O antigen typing. The sentence "consisting of 26 of the 30 APEC strains" has been deleted as irrelevant and confusing.**

Line 30: Again, please be more specific, what kind of enzymes are we talking about? Are these lysins or are the enzymatic activity part of a receptor binding protein?

- **It is an enzymatic activity that is part of a receptor binding protein, the precision was provided.**

Line 30-31: It's a bit weird mentioning the REC phage here, I would suggest that you explain this before going to the host range, so move to line 29.

- **Modified as suggested.**

Line 52: Add "to survive the APEC infection". Delete "we think that this"

- **Added and deleted as suggested (line 46).**

Introduction

Lines 72 to 76: It would be relevant to explain more about what are these phages? Taxonomy is changing so if the genomes are out there, please use the current taxonomy and explain more about these phages. Are they o-antigen specific or have other receptors?

- **The taxonomy of phage R has been added (line 65). Genome of phages SPR02 and TM3 are not available in the databases. Moreover, in the cited articles, there is no data of their putative receptors. We indicated that the two phages belong to unknown genus (lines 69 and 71) and we removed “siphovirus” when phage TM3 was cited.**

Lines 84-86: It is not important that the phages are morphological distinct, more important is their host range and receptors as well as ability to overcome anti-phage defense systems of the host.

- **"Morphologically" has been deleted. As suggested, we added that important features for the construction of a phage cocktail are the fact that phages are able to target different bacterial receptors and to overcome bacterial anti-phage defense systems resistance, as well of having a wide host range (line 80).**

Lines 88-92: Again, relevant to add information about what are these phages? It is not particular relevant if they are myoviruses, but the about mentioned biological features are important as well as genera if the sequences are known.

- **Except for the phage G28 (a *Tequatrovirus*) in the cocktail, none of the other phages mentioned in Korf *et al* study (reference no. 12) have been sequenced. No more information concerning these phages were provided. Terms referring to the old taxonomy (myovirus and siphovirus) have been removed (line 85).**

Lines 93: Here you could state more strongly your aim.

- **According to your suggestion, the aim of our study has been added (line 90).**

Line 95: All phages were new, but one phage did only show 78% identity to a phage in the database, and therefore you can define a new genus. Please revise.

- **The sentence was revised to more clearly indicate that a new genus was defined.**

Line 97: The phages do not contain polysaccharides, please clarify if this is an enzymatic activity as part of the RBP?

- **The sentence has been clarified (line 99).**

Results and discussion

Please combine the three first paragraphs into one that focus on the new findings and describes the collection in a short and precise way.

- **According to your suggestion, the three paragraphs were combined into one (“Isolated coliphages belong to nine genera”). The phage collection was described in a short and more focus way.**

Line 105-106: What is this used for? It seems irrelevant here. Please delete.

- **This information has been deleted.**

Line 108: Please add the RFLP data as supplementary.

- **RLFP data have been added as an additional figure (Fig S1). Among the 66 isolated phages, genome restriction by HincII was performed for 42 phages. Accordingly, we corrected this line 106. Method was also corrected (lines 372 to 374). It should be noted, that the genome of three phages (*Dhakavirus* ESCO10, *Mosigvirus* ESCO47 and *Tequatrovirus* ESCO58) could not be restricted by HincII (neither by HaeIII). Hypothesizing that there could be different phages, their genome was sequenced leading to three additional groups.**

Line 109-113: It is weird to introduce the REC phage before the two ancestors. Please move to later.

- **This paragraph has been moved to the end of the part, to introduce the parental phages before.**

Line 115-127: this data can be added into a table and shown either as supplementary or as part of one of the other tables. This information is nice, but not something to comment on, unless you see something new to say.

- **Information on the diameters of the heads and tails of 19 phages was added in Supplementary Table 1.**

Lines 124-127: This is not important and refer to old taxonomy, please delete this.

- **Deleted, as suggested.**

Lines 129-131: This can be seen from the tables and can be deleted.

- **This information was deleted.**

Line 132-134: this could be your starting point.

- **We found it clearer to start by describing shortly that based on RFLP analysis, 18 phages were selected to be sequenced among the 42 isolated phages. Then, information on which genus they belonged was provided.**

Line 134-138: Delete as part of the table.

- **The nine genera were still indicated in the text. However, this paragraph was shortened by deleting the phage names.**

Line 154-166: This is relevant and could be moved to line 134, where you refer to figure S2. And then follow by the parts in line 140-146.

- **Modified as suggested. According to the editor's suggestion, the tree (Figure 2) and the analysis were modified to be more relevant.**

The paragraph starting on line 167 concerning the genomes could be shortened and focused on the synteny as the title says. In this paragraph you also present differences in gene content of

the phages in each genus. But the question is why is this relevant for your study? Please clarify.

- **We agree that the differences presented were not relevant in the original manuscript. We then focused on differences in structural genes that could have an impact on the host range.**

Line 168-182: I would suggest deleting this, as it is not important for the aim and nothing new is found.

- **Indeed, this information can be found in Table 2, it was deleted. The information on the ESCO41 *Nouzillyvirus* phage has been updated (lines 124 and 154). We would like to still mention the *Nouzillyvirus* protein which have no homolog in databases (yet) (line 153).**

Line 191- 205: How can you demonstrate the recombination spot with data and in a figure?

- **We carefully checked the identity between the genomes of the parental phages ESCO05 and ESCO37 with the genome of the recombinant phage, REC. Contrary to what we have hypothesized, the identified inverted repeat could not correspond to the recombination site since the right inverted repeat is not located in a conserved region between the phages. Consequently, we remove this hypothesis in the revised version of the manuscript.**

Line 206-209: Move to before explaining the REC phage, so you introduce the parental phages first.

- **According to your suggestion, sentences were moved to introduce the parental phages before describing the REC phage (line 164 to 168).**

In the next paragraph starting on line 227, I miss the rational for focusing on lysins and spannins? Why are they specifically analyzed? I don't see the relevance for this work. In any case, all genes discussed and analyzed should be mentioned by name, and these should appear in the table as well.

- **The part on the analysis of the enzymes involved in bacterial lysis in the late steps of the phage life cycle was deleted. We agree that these data were not relevant for this study. We then focused on enzymes that could facilitate the adsorption step. The names of all genes discussed in the manuscript were mentioned.**

Line 249-258: not relevant and can be deleted.

- **This part was deleted.**

I think the manuscript would benefit from combining analysis of the adsorption genes with the host range part.

- **Thanks to your comment, as the two parts were correlated, the two analyses were combined.**

Line 264 and onwards: It would be nice to mention the genus of the phages while presenting and discussing their host range as this can direct you to the potential receptors of the phages. This could also help you relating to relevant literature which I think could provide a more thorough discussion.

- **As suggested, the genus of the phages was mentioned. We strengthened the discussion by referring more to relevant literature and by focusing on potential receptors and phage-associated enzymes.**

Line 269: Tequintavirus do not have many tail fibers, they have three lateral tail fibers and a central spike, recognizing the o-antigen and an outer membrane protein as receptor, respectively. Please rephrase.

- **Thank you for the correction of this error, the sentence was rephrased (line 217).**

Line 272-274: It is not that relevant what others found in terms of numbers of strains infected, as these collections are so different from each other. I find it more relevant to discuss the genus/RBPs/host range.

- **Indeed, the collections are very diverse and sometimes without information on the serotypes of the strains. This was corrected.**

Line 275-278: Please relate to genus and putative RBPs.

- **Modified as suggested.**

Line 279-280: I respectfully disagree. For therapy, only infection is important and the following parts are not relevant. Lysis without infection is a dead end for the phages, and no propagation will take place, thus omitting the positive effect of auto-replication of the antimicrobial activity. Also, lysis without infection as an antibacterial strategy will requires a high level of phages applied and this is simply not feasible.

- **We agree that lysis without infection is a dead end for phages. Nevertheless, it may give an indication of potential enzymatic activities. However, this paragraph has been deleted.**

Line 285-287: What elements? Yes, the depolymerases do have important role in infection, please elaborate on their function as RBPs.

- **We agree with you that the sentence “while other elements” was not precise. This sentence was deleted. We more clearly indicate that these depolymerases degrade capsule and might allowed phages to reach their bacterial receptor (line 244).**

I like the in ovo experiment, but still I'm not convinced that including phages that do not infect the pathogenic APEC is a good idea. Did you isolate any phages after the experiment? And if yes, what phages was present?

- **That's a very pertinent remark. Indeed, only 4 of the 8 phages can multiply on BEN4358. However, our aim was to build a cocktail with phages targeting various APEC strains, justifying the inclusion of the 4 phage genera which do not lyse the strain used.**

Six days post inoculation, allantoic fluid were sampled. Phages titers were between 10^3 and 10^9 PFU/mL. A PCR was performed on only one sample and ESCO9, REC, ESCO3, ESCO47 and ESCO58 were detected. As this was done on only one sample, we did not present this data in the manuscript.

April 6, 2023

Dr. Catherine Schouler
Institut National de Recherche pour l'Agriculture l'Alimentation et l'Environnement
INRAE, Université de Tours, ISP
Nouzilly 37380
France

Re: Spectrum04296-22R1 (Isolation and characterization of a novel phage collection against avian pathogenic Escherichia coli)

Dear Dr. Catherine Schouler:

There are only few minor comments left to address. If you can respond to these I should be able to quickly accept this manuscript.

Link Not Available

Sincerely,

Thomas Denes

Journals Department
Reviewer comments:

Reviewer #1 (Comments for the Author):

This revised manuscript has improved greatly and is now well structured and to the point. It demonstrate the road from phage isolation to applications. I only have a few comments:

Line 26: add "different" after "nine"

Line 28: at the end of the sentence add "isolated in this study"

Line 29: Delete "the"

Line 32: sentence can be combined with the following sentence into something like this: "To demonstrate the therapeutic

potential, a phage cocktail consisting of eight phages belonging to different genera was tested against BEN4358, a pathogenic APEC O2 strain."

Line 42: correct increase to increased

Line 43: Delete "strains"

Line 80: I think "construction" should rather be "design" or "development"

Line 90: Maybe add what is the research gap here, something like "However, no studies..."

Line 226: T5 like phages may recognise other receptors that FhuA, you mention this yourselves in line 247. I suggests that you add something about this here.

Line 271: I suggest that you add an explanation for why you also included phages in the cocktail that are not infecting the strain used.

Line 276: Add how many hours the experiment ran. Its relevant for the reader to have this clearly said.

Line 292: I would delete the "specific" as you already said that phages are specific.

Staff Comments:

Preparing Revision Guidelines

Please return the manuscript within 60 days; if you cannot complete the modification within this time period, please contact me. If you do not wish to modify the manuscript and prefer to submit it to another journal, please notify me of your decision immediately so that the manuscript may be formally withdrawn from consideration by Microbiology Spectrum.

Reply to reviewer's comments:

Line 26: add "different" after "nine"

- **Added as suggested (line 26).**

Line 28: at the end of the sentence add "isolated in this study"

- **This information was added at the end of the sentence (line 28).**

Line 29: Delete "the"

- **Deleted as suggested (line 29).**

Line 32: sentence can be combined with the following sentence into something like this: "To demonstrate the therapeutic potential, a phage cocktail consisting of eight phages belonging to different genera was tested against BEN4358, a pathogenic APEC O2 strain."

- **The two sentences were combined as follows: "To demonstrate the therapeutic potential, a phage cocktail consisting of eight phages belonging to eight different genera was tested against BEN4358, an APEC O2 strain" (line 32).**

Line 42: correct increase to increased

- **Correction has been done (line 41).**

Line 43: Delete "strains"

- **Deleted as suggested (line 42).**

Line 80: I think "construction" should rather be "design" or "development"

- **"construction" was replaced by "design" (line 79).**

Line 90: Maybe add what is the research gap here, something like "However, no studies..."

- **As suggested, we add the research gap to precise the aim of the study: "However, no studies have combined both genomic characterization of a collection of phages against APEC strains with an *in vivo* evaluation of their therapeutic potential to control avian colibacillosis" (line 89).**

Line 226: T5 like phages may recognise other receptors than FhuA, you mention this yourselves in line 247. I suggest that you add something about this here.

- **According to your suggestion, it was indicated that *Tequintaviruses* also recognize BtuB or FepA (line 223).**

Line 271: I suggest that you add an explanation for why you also included phages in the cocktail that are not infecting the strain used.

- **We aimed to design a phage combination that took into account the diversity of our phage collection. We added this explanation line 264. We slightly modified the sentence lines 270 to 273.**

Line 276: Add how many hours the experiment ran. It's relevant for the reader to have this clearly said.

- **To be clearer, the sentence was modified as follows: “The phage cocktail treatment allowed 90% of the chicken embryos to survive a 6-day infection by the BEN4358 strain, in contrast with the non-phage treated bacterial control group, in which all the embryos had died at day 4” (line 280).**

Line 292: I would delete the "specific" as you already said that phages are specific.

- **The word “specific” was deleted as suggested, to avoid repetition (line 292).**

April 12, 2023

Dr. Catherine Schouler
Institut National de Recherche pour l'Agriculture l'Alimentation et l'Environnement
INRAE, Université de Tours, ISP
Nouzilly 37380
France

Re: Spectrum04296-22R2 (Isolation and characterization of a novel phage collection against avian pathogenic Escherichia coli)

Dear Dr. Catherine Schouler:

Your manuscript has been accepted, and I am forwarding it to the ASM Journals Department for publication. You will be notified when your proofs are ready to be viewed.

Sincerely,

Thomas Denes
Editor, Microbiology Spectrum
